# Immunopathological Changes Caused by *Oesophagostomum radiatum* in Calves: Insights into Host–Parasite Interactions

**DOI:** 10.3390/pathogens14111074

**Published:** 2025-10-22

**Authors:** Cesar Cristiano Bassetto, Ana Cláudia Alexandre de Albuquerque, José Gabriel Gonçalves Lins, Guilherme Fernandes Dias Canalli, Anandra Kauára dos Santos Gomes, Alessandro Francisco Talamini Amarante

**Affiliations:** 1Department of Biodiversity and Biostatistics, Institute of Biosciences, São Paulo State University (UNESP), Botucatu 18618-689, SP, Brazil; alessandro.amarante@unesp.br; 2Department of Pathology, Reproduction, and One Health, School of Agricultural and Veterinary Sciences, São Paulo State University (UNESP), Jaboticabal 14884-900, SP, Brazil; claudia.albuquerque@unesp.br (A.C.A.d.A.); guilherme.canalli@unesp.br (G.F.D.C.); kauara.anandra@gmail.com (A.K.d.S.G.); 3Department of Veterinary Clinics, School of Veterinary Medicine and Animal Science, São Paulo State University (UNESP), Botucatu 18618-681, SP, Brazil; jose.lins@unesp.br

**Keywords:** gastrointestinal nematode, cattle, ruminants, parasites, immunity

## Abstract

The intensity and prevalence of different gastrointestinal nematode species vary across regions worldwide. *Oesophagostomum radiatum* commonly shows a high occurrence in young cattle. *O. radiatum* causes anaemia, hypoproteinaemia, and immunopathological changes in the large intestine wall, impairing calves’ body weight gain. This study aimed to assess the impact of natural *O. radiatum* infection on haematological parameters and immune responses in 23 Nellore calves, considering sex-based differences. Assessments included *Oesophagostomum* egg count (EPG), worm count, packed cell volume (PCV), total plasma protein, histopathological and immunohistochemistry analyses. A large number of parasites attached to the colon mucosa were observed, along with massive nodule formation and haemorrhagic lesions, mainly within a 20–30 cm-long segment adjacent to the nodules. The maximum mean egg shedding was approximately 165 EPG for males and 173 EPG for female calves; however, males presented a significantly higher worm count (969 ± 200.5) than females (460 ± 99.5). There were significant positive correlations between the total *O. radiatum* worm count and *O. radiatum* EPG for both female and male calves. Significant negative correlations were observed between the total *O. radiatum* worm count and PCV in female calves. Our results demonstrated that natural *O. radiatum* infection in Nellore calves induced marked immunopathological alterations, including chronic inflammatory responses that impaired intestinal function. Sex-related differences suggested that female calves may develop more effective tissue responses. These findings emphasise the economic impact of subclinical infections and reinforce the importance of control strategies to minimise productivity losses in cattle.

## 1. Introduction

Several gastrointestinal nematode species parasitise cattle worldwide. The most prevalent species in Brazil are *Haemonchus placei*, *Cooperia punctata*, and *Oesophagostomum radiatum* [1,2]. In Minas Gerais State, Brazil, *Cooperia* sp. represented 74.4% of the total nematodes recovered from Zebu cattle, followed by *Haemonchus* sp. (19.2%) and *Oesophagostomum* sp. (4.5%) [3]. Despite these relatively low numbers compared with *Cooperia* and *Haemonchus*, *O. radiatum* commonly shows a high occurrence in young cattle. Epidemiological studies conducted in São Paulo and Minas Gerais states, Brazil, reported prevalence rates of 100% and 95%, with average infection intensities of 579 and 471 *O. radiatum* worms, respectively [4,5].

Regarding the life cycle of *O. radiatum*, studies in naïve calves experimentally infected with a single dose of infective third-stage larvae (L3) have shown that the third moult occurs within cysts located in the submucosa of the ileum and colon, ranging from 5–7 days post-infection [6] to 7–9 days post-infection [7]. Most fourth-stage larvae (L4) migrated from the cysts into the intestinal lumen after day 10 [7], where the fourth moult occurred between 17 and 22 days [6]. The prepatent period ranged from 32 to 42 days, and egg production usually peaked between the sixth and tenth weeks [6]. The rate of establishment and development of the larvae is influenced by host age. Six weeks after infection with 5000 L3, an average of 403 and 1174 adult worms were recovered from naïve animals aged 13 and three months, respectively [8].

Due to its mucosal migration and hematophagous behaviour, *O. radiatum* causes haemorrhagic ulcerated lesions in the large intestine mucosa. The synergism of its pathogenic mechanisms results in faecal blood loss, anaemia, and hypoproteinaemia in infected animals, impairing body weight gain in young cattle. Alterations in gastrointestinal motility and digesta flow may partially explain the reduced appetite and are also associated with the occurrence of diarrhoea [9,10,11,12]. In detail, Elek and Durie [13] described the immunopathological changes in the intestinal wall of previously infected animals. According to these authors, the elimination of L3 18–22 h post-reinfection correlated with oedema, hyperaemia, and hypersecretion in the intestine, subsequently leading to brief yet severe diarrhoea. Additionally, *O. radiatum* moulting to the fourth stage was correlated with recurrence of oedema and vasculitis, which led to the isolation and elimination of larvae together with adjacent necrotic host tissue. When larvae were not expelled, the healing process was prolonged, and the nodular lesion remained. Diarrhoea noted following the expulsion of fourth-stage larvae was linked to gut wall oedema, mucosal gland hypertrophy, and excessive secretion [12]. Irregular and fibrotic nodules spread in the large intestine, caused by *Oesophagostomum* infection, reduce the suitability of these tissues for food production [14,15].

Prophylaxis of gastrointestinal nematodes infections has been mainly based on the use of anthelmintics; however, the emergence of nematode populations resistant to all commercially available products has become a major concern in cattle farming. Several reports describe *O. radiatum* populations with anthelmintic resistance, reviewed by [16].

A study with Nellore calves showed that reductions of 20% and 28% in the body weight gain were caused primarily by *O. radiatum* infection after weaning [17]. Given the great importance of oesophagostomosis in cattle, the objective of this study was to evaluate and characterise the effects of a natural *O. radiatum* infection on haematological parameters and immune and inflammatory responses in Nellore calves, with attention to potential sex-based differences, to better understand host–parasite interactions and associated tissue responses.

## 2. Materials and Methods

### 2.1. Animals and Management

All procedures adhered to the Animal Welfare Standards, and the study received approval from the Institutional Ethics Committee of the Institute of Biosciences (protocol code 262-CEEA, approved on 3 December 2010).

From 29 January to 15 March 2011, 23 Nellore calves, 11 females and 12 males, were born and reared on *Urochloa brizantha* cv. Marandu permanent pasture at the experimental cattle farm of São Paulo State University (UNESP) in Botucatu, Brazil (22°81′ S, 48°40′ W; 554 m above sea level).

Cows and calves grazed together throughout the experiment in 14 paddocks (1.5 ha each), with free access to water and mineral salt. When pasture availability was insufficient, the animals were relocated to a new paddock.

After weaning, when cows were transferred to another farm and calves remained in the same paddocks (15 September 2011), the calves received concentrate supplementation (800 g per calf; Geramilk Bezerra^®^, Presence, Descalvado, São Paulo, Brazil) mixed with a coccidiostat (decoquinate 0.5 mg/kg, Deccox^®^, Alpharma, Piscataway, NJ, USA) from 15 September until 15 November 2011 and continued receiving concentrate alone until 15 December 2011.

Health management comprised a clostridial vaccination on 7 July 2011 (Carbun-vet Polivalente^®^, Biovet, Vargem Grande Paulista, São Paulo, Brazil) and a Foot and Mouth Disease immunisation on 10 November 2011 (Aftobov Oleosa^®^, Merial, Campinas, São Paulo, Brazil). To control ectoparasites, mainly *Haematobia irritans*, calves received cypermethrin (0.5 g/100 kg, Cypermil^®^ Pour On, Ouro Fino, Cravinhos, São Paulo, Brazil). On 28 July 2011 (day 0), all calves were drenched with albendazole (10 mg/kg; Endazol 10% Co^®^, Irfa, Ilhéus, Bahia, Brazil) to eliminate any pre-existing gastrointestinal nematodes and allow for uniform reinfection.

All calves were slaughtered at an abattoir on 15 March 2012, at an average age of 13 months.

### 2.2. Haematological and Parasitological Procedures

Blood samples were obtained from the animals via jugular venipuncture utilising tubes containing EDTA (Vacutainer^®^, BD, Franklin Lakes, NJ, USA). These samples were used to ascertain packed cell volume (PCV) and total plasma protein (TPP) using microcentrifugation and refractometry, respectively.

Faecal samples were obtained directly from the rectum of animals to ascertain nematode egg counts (FEC) and to identify infective larvae (L3) genus. A modified McMaster method was employed, where each counted egg corresponded to 50 eggs per gram of faeces (EPG), after that faecal cultures were used for L3 identification [18]. The *O. radiatum* FEC was estimated based on the EPG values and L3 percentage from faecal cultures.

After slaughter, the abomasum, small intestine, and large intestine were preserved at −20 °C until processing for worm recovery. After thawing at room temperature, the large intestine was washed separately with water, and a 10% subsample was collected and examined. All worms adhering to the mucosa of the large intestine were collected and counted. Worms were quantified, sexed, and categorised according to their developmental stage and species, then stored in 70% ethanol [18,19]. In addition, tissue samples were collected, fixed in 4% formalin, and kept in 70% ethanol until histological procedures.

### 2.3. Histological Procedures

Formalin-fixed and paraffin-embedded large intestine colon tissues were cut to 5 μm thickness, followed by dehydration and staining with haematoxylin and eosin or toluidine blue (1%) for histopathological analyses and eosinophil and mast cell counts, respectively. Eosinophils and mast cells were counted at a magnification of ×1000, using a 1 cm^2^ eyepiece graticule, within 30 randomly selected fields of view, and the results were reported as the number of cells per mm^2^. In addition, the mucosal thickness was measured in two randomly selected fields of the intestinal mucosa for each animal, using a magnification of ×100, and the resulting measurements were expressed in micrometres (µm) [20].

### 2.4. Immunohistochemistry

The immunohistochemistry technique was performed based on the method described in [21], with a few modifications. Formalin-fixed and paraffin-embedded large intestine colon tissues were cut to 4 μm thick, mounted on charged glass slides (Starfrost Advanced Adhesive, Knittel^®^, Braunschweig, Germany), and rehydrated. Following the heat-induced antigen retrieval in 10 mM citrate buffer at pH 6.0 in a steamer at 95 °C for 40 min. After cooling the slides, they were washed with distilled water and incubated in 0.3% hydrogen peroxide for 15 min at room temperature (RT) to quench endogenous peroxidase activity. After washing with EnVision™ FLEX Wash Buffer (20×) (Dako^®^, Carpinteria, CA, USA), the tissue was soaked in a 10% non-fat dry milk solution for 15 min at RT to prevent hydrophobic background staining. Primary antibodies were diluted in EnVision™ FLEX antibody diluent (Dako^®^) and incubated for 60 min at RT with polyclonal rabbit anti-human CD3 (clone: A0452, Dako; ready to use) or monoclonal mouse anti-human anti-HLA-DR (clone TAL.1B5, Dako, 1:500 dilution). Slides were washed in PBS pH 7.2, and the appropriate secondary antibody was applied to sections for 30 min at RT. After a PBS pH 7.2 wash, sections were incubated with 3,3′-diaminobenzidine (DAB) for 1′30″ at RT, washed in distilled water, counterstained with haematoxylin, dehydrated, and mounted in Erv-mount (Easypath^®^, Indaiatuba, São Paulo, Brazil). Lymph nodes from a healthy sheep were used as positive controls, and for negative controls, and no primary antibodies were added.

Positive cell counts were performed in 10 random areas of each slide using a 1 cm^2^ eyepiece graticule at ×400 magnification under an optical microscope and expressed as the number of positive cells per mm^2^.

### 2.5. Procedure of Scanning Electron Microscopy (SEM)

Tissue samples from the large intestine colon, containing parasites, were collected and fixed in 2.5% glutaraldehyde solution diluted in phosphate buffer (pH 7.3–0.1 M) for a duration of 48 h at room temperature. The samples were subsequently rinsed with distilled water and post-fixed in 1% osmium tetroxide diluted in distilled water for 30 min at room temperature. After fixation, the samples underwent dehydration through a series of ethanol solutions, were dried using critical point drying with CO_2,_ and were sputter coated with gold (Bal-Tec SCD 050). The specimens were examined using a Quanta 200 scanning electron microscope (FEI Company, Eindhoven, The Netherlands) at an accelerating voltage of 12.5 kV.

### 2.6. Statistical Analyses

Data were submitted to a normality test (Shapiro–Wilk test), which revealed that the final live weight, TPP, PCV, eosinophil count, mast cell count, mucosal thickness, and MHC II+ cell count followed a normal distribution. To investigate potential statistical differences between male and female calves, we employed either an unpaired *t*-test for normally distributed data or a Mann–Whitney test for non-normally distributed data. Moreover, since the *O. radiatum* worm count did not follow a normal distribution, variables were correlated and analysed using Spearman’s correlation and linear regression using GraphPad Prism version 8.2.1. Results are presented as mean values ± standard error of the mean.

## 3. Results

*Oesophagostomum radiatum* adults present very well-developed cephalic vesicles, lateral alae, and cervical papillae. They have a small buccal capsule, with the oral aperture surrounded by the internal and external leaves of the corona radiata. The cephalic vesicle surrounds the buccal capsule and is separated from the cephalic alae by an annular constriction, while a cervical groove separates the cephalic alae from the lateral alae (Figure 1A and Figure 2A). The cephalic alae exhibit an annular depression in their posterior region. Two well-pronounced cervical papillae were observed right after the cervical groove. Males displayed a bell-shaped bursa with two lateral lobes, two long-shaped spicules with similar length, and a gubernaculum (Figure 1B).

Electron microscopy revealed the presence of tunnels on the mucosal surface (Figure 2B), some of which contained embedded parasite.

A light *Oesophagostomum* sp. infection was observed at the first sampling (late July), which was eliminated following anthelmintic treatment. Then, a new patent *Oesophagostomum* infection was detected from October onwards, with the highest FECs recorded between December and March, with a peak on 2nd February, with male calves shedding about 165 EPG and females 173 EPG, when the calves were approximately 10 to 13 months of age (Figure 3, Appendix A).

All calves were infected by *O. radiatum*, with males displaying significantly (*p* < 0.05) higher mean infection intensity (969 ± 200.5) than females (460 ± 99.5) (Table 1 and Appendix A). The minimum and maximum *Oesophagostomum* worm burdens were, respectively, 280 and 2363 worms for males and 50 and 1016 worms for females. Females exhibited higher mean values of the colon mucosal thickness than males (*p* = 0.0326). A very low burden of *Trichuris* was observed in the large intestine of the calves.

Regardless of sex, there was a significant positive correlation between the total *O. radiatum* count and *O. radiatum* FEC (r = 0.8003 and R^2^ = 0.5992; *p* < 0.0001) (Figure 4A) and a significant negative correlation between total *O. radiatum* worm count and PCV (r = −0.6683 and R^2^ = 0.3728; *p* = 0.0005) (Figure 4C). Furthermore, there were no significant correlations between total *O. radiatum* burden and TPP (Figure 4B), final live weight (Figure 4D), eosinophil (Figure 4F), mast cell (Figure 4G), CD3+ cells (Figure 4H), MHC II+ cells count (Figure 4I), mucosal thickness (Figure 4E), nor *Trichuris* burden (Figure 4J). A similar trend was observed regarding sex, with no significant correlation between *O. radiatum* burden and effector and immunolabelled cells, mucosal thickness, and *Trichuris* burden.

Nonetheless, we identified significant positive correlations between the total *O. radiatum* worm count and *O. radiatum* FEC for female (r = 0.7563 and R^2^ = 0.5757; *p* = 0.0093) (Figure 5A) and male (r = 0.8386 and R^2^ = 0.7343; *p* = 0.0011) (Figure 5D) calves. Additionally, significant negative correlations were observed between the total *O. radiatum* worm count and PCV in female calves (r = −0.7945, R^2^ = 0.6252; *p* = 0.005) (Figure 5C).

In infected animals, a large number of parasites were observed attached to the colon mucosa, accompanied by massive nodule formation along the large intestine (Figure 6). Haemorrhagic lesions were observed, mainly within a segment about 20–30 cm long on the colon, adjacent to these nodules (Figure 7).

Histopathological analyses (Appendix A) revealed extensive tissue damage with some irregularities, including disruption of normal architecture and haemorrhagic lesions, with multiple parasites present in the lumen (Figure 8A), lamina propria of mucosa (Figure 8B,C), and submucosa (Figure 8D) of the large intestine. A marked and diffuse mixed inflammatory infiltration, primarily consisting of lymphocytes, plasma cells, and eosinophils, was observed in the mucosa (Figure 8C). The same cell infiltrate was observed to have diffusely spread to the submucosa, ranging from discrete to marked infiltration (Figure 8D). Moderate mucosal fibrosis with discrete follicular hyperplasia was identified in some animals. The inflammatory infiltrate also caused focal destruction of the internal muscularis, moderately extending into the submucosa (Figure 8D). Furthermore, the superficial epithelium of the intestinal villi showed moderate detachment toward the lumen (Figure 8B), with multiple foci of follicular hyperplasia of varying sizes.

Interestingly, an animal with one of the lowest EPG and worm count values exhibited a focally extensive mucosal area containing a high amount of basophilic and refringent material, suggestive of calcification (Figure 8D), surrounded by prominent fibrocollagenous tissue rich in lymphocytes, plasma cells, and occasional macrophages. Moreover, all parasites found in the mucosa and submucosa were surrounded by a significant number of eosinophils, plasma cells, and cellular debris, which were enclosed by prominent fibrocollagenous tissue rich in lymphocytes, plasma cells, and eosinophils located near the submucosa. Some nodules lacking visible parasites exhibited similar histological features (Figure 8E).

Additionally, we observed the presence of mast cells close to immature stages, although they were less abundant than eosinophils. A larval stage was observed in the deep mucosal layer of the large intestine mucosa, with its buccal capsule filled with epithelial cells (Figure 8A). Immunolabelled CD3+ and MHC II+ cells were widely distributed throughout all intestinal mucosal layers, with a marked presence in the lamina propria, while CD3^+^ cells were also present intraepithelially (Figure 9A,B,D,E, Appendix A).

## 4. Discussion

This study demonstrated the occurrence of severe alterations in the mucosal surface of the large intestine of Nellore calves naturally infected with *O. radiatum*. Macroscopic lesions were mainly located in the proximal colon, in agreement with the descriptions of [12], who reported lesions in the proximal 61 cm of the colon after the conclusion of the histotrophic phase of larval development.

The *O. radiatum* adults are usually present in the lumen of the large intestine, attached to the mucosa [22]. Their hematophagous habit and buccal morphology result in extensive tissue damage and haemorrhagic ulcerated lesions, associated with mucosal thickening and visible oedema, as observed in *O. columbianum* in sheep [23] and goats [22], and *O. dentatum* in swine [24]. We could observe, during the histotrophic phase, that immature stages of *Oesophagostomum* sp. feed on mucosal tissue, which may contribute to the tissue destruction. Their presence and feeding habits elicit a strong local immune response involving lymphocytes, plasma cells, eosinophils, and mast cells, accompanied by degeneration of epithelial cells [24].

The pronounced fibrocollagenous tissue with mixed inflammatory infiltrate around *Oesophagostomum* larvae is characteristic of granuloma formation [7]. A vimentin-stained parasitic nodule from *Oesophagostomum*-infected sheep showed a diffuse fibrous connective tissue capsule surrounding the necrotic and calcified region with larval stages [23]. A T helper 2 (Th2) type adaptive immune response is typically elicited against gastrointestinal nematodes, involving cellular, cytokine, and humoral components [25]. T lymphocytes (CD3+), dendritic cells, macrophages and B lymphocytes (MHC II+) are present in the Th2 response and are central to this chronic process [26]. A strong and chronic immunological response against the histotrophic phase larvae reflects the host’s attempt to isolate and eliminate infection [12].

Because of the lesions caused in the intestine, diarrhoea was the main clinical manifestation of oesophagostomosis in the animals evaluated in the present experiment. Considering the faecal consistency measurement using a 4-level scoring scale: 0 = normal (firm but not hard); 1 = soft (does not hold form, piles but spreads slightly); 2 = runny (spreads readily); and 3 = watery (liquid consistency, splatters) [27]. We observed that the animals frequently presented runny faeces, particularly in the last three months of the trial. In addition to damage caused by established parasites, continuous ingestion of infective L3 from pasture contributed to disease pathophysiology. According to [12], the elimination of third-stage larvae 18–22 h post-administration of the reinfecting dosage from a sensitised host was correlated with oedema, hyperaemia, and hypersecretion in the gut, subsequently leading to a brief yet severe episode of diarrhoea, potentially resulting in anorexia and weight loss. The present study did not record the severe clinical cases of oesophagostomosis mentioned by [12], which included anorexia and weight loss. The relative protection was probably due to the elicitation of an immune response resulting from the continuous ingestion of infective *O. radiatum* larvae. However, these subclinical infections can be a significant cause of reduced productive performance in young cattle. Compared to a group treated monthly with an anthelmintic effective against *O. radiatum*, animals in the control group, without monthly preventive treatment, showed a reduction in weight gain ranging from 20% to 28% [17]. Possibly, the losses in performance, in the case of natural infections, result not only from the mechanical damage caused by the adult worms but also from the lesions produced in the mucosa during the histotrophic phase of development, as demonstrated in Figure 4, aggravated by immunopathological changes leading to granulomatous nodules.

Our results about the influence of the animal sex on the susceptibility agree with other studies that also demonstrated higher susceptibility of males to helminths in comparison with females of other species of mammals [28]. Possibly, for this reason, the depression of body weight gain is more severe in males than in female Nellore calves infected with *O. radiatum* [17]. Females had greater mucosal thickness values (826.4 µm) than males (743.5 µm), indicating that the tissue hyperplasia may play a role in the protection of females against *Oesophagostomum* infection. Although both male and female calves showed low PCV and negative correlation between *O. radiatum* worm count and PCV, only females had a significant correlation. This observation could be attributed that females have a more intense tissue and immune response that could impair parasite establishment, but this robust response has an energetic cost. This duality suggests that PCV in females is not only attributed to parasite-induced blood loss but also immunopathological response.

The present study demonstrated that the *Oesophagostomum* EPG count, estimated based on the proportion of infective larvae obtained from faecal cultures, showed a significant correlation (r = 0.78) with the parasite worm burden. A study with young cattle conducted in Australia also demonstrated that the shedding of *O. radiatum* eggs is associated with the number of adult females [29]. Correlation coefficients between worm burden and egg count varied between 0.91 and 0.97 in seven trials with 84 Friesian bull calves given a single challenge infection of *O. radiatum* [30]. Taken together, this information confirms that FECs associated with larval differentiation are reliable methods to estimate the worm burden of *Oesophagostomum*.

Correlations between EPG × body weight and worm burden × body weight were low and non-significant. In the present study, the highest mean *Oesophagostomum* egg count was on 2nd February, with males shedding about 165 EPG and females 173 EPG. This EPG value was lower than those found by [9] in calves 3–4 months old and experimentally infected with 10,000 L3, which showed the maximum individual FEC ranged from 680 to 2490 EPG.

There was an inverse relationship between the intensity of the infection and the haematological variables analysed (PCV and TPP), demonstrating the occurrence of losses due to feeding activities of the parasites that are associated with tissue damage and immunopathological changes. Calves 3–4 months old infected experimentally with 10,000 L3 presented a fall of serum protein concentrations from a mean pre-infection level of 7.9 g/dL to a mean minimum value of 4.0 g/dL during the sixth week after infection [9]. Bleeding into the parasitised colon began 3 weeks after infection with 7000 L3, at the time of the fourth larval moulting, and persisted for at least 9 weeks. Maximum erythrocyte losses (mean 39 mL packed erythrocytes/day) occurred during the seventh week after infection and were associated with a mean worm burden of 2500 adult parasites [10]. The colon was the major site of blood loss, frequently in association with diarrhoea [10]. Therefore, intestinal leakage of plasma protein and bleeding might be considered the principal factors causing reduction in TPP and PCV in calves naturally infected with *O. radiatum*.

## 5. Conclusions

In conclusion, natural *O. radiatum* infection in Nellore calves induced marked immunopathological alterations, with chronic inflammatory responses that impair intestinal function and contribute to reduced weight gain, even in the absence of severe clinical signs. The reliability of faecal egg counts was confirmed by their correlation with worm burden, validating their usefulness for estimating infection levels in cattle. Moreover, sex-related differences suggest that females may develop more effective tissue responses, conferring partial protection against parasite-induced damage. These findings emphasise the economic impact of subclinical infections and reinforce the importance of control strategies to minimise productivity losses in cattle.

## Figures and Tables

**Figure 1 pathogens-14-01074-f001:**
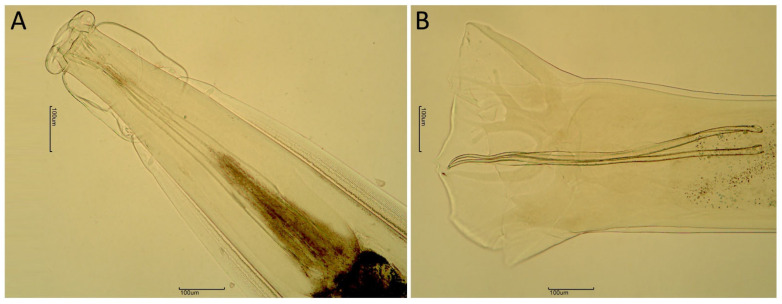
Details of *Oesophagostomum radiatum* morphology. (**A**) Anterior part of *O. radiatum* with pronounced cephalic vesicle, cephalic alae and lateral alae; (**B**) Posterior part of *O. radiatum* showing the long and tapered-shaped spicules with bell-shaped bursa.

**Figure 2 pathogens-14-01074-f002:**
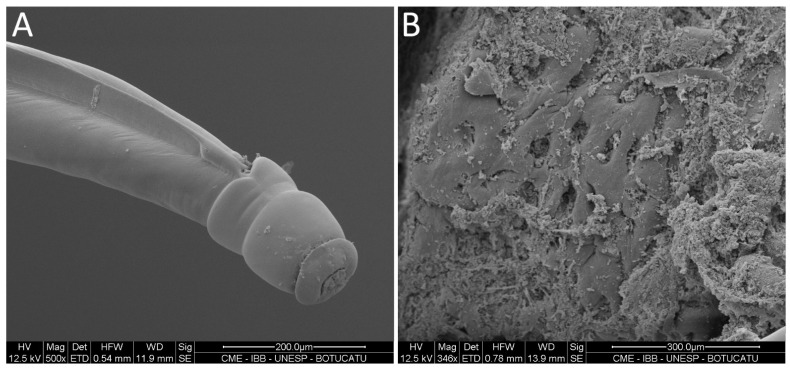
(**A**) Details of the anterior part of *Oesophagostomum radiatum* morphology; (**B**) Tunnels on the mucosal surface of the large intestine of a calf caused by the parasite.

**Figure 3 pathogens-14-01074-f003:**
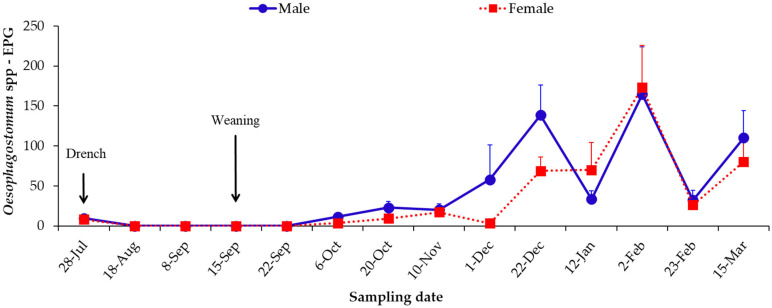
Mean number of *Oesophagostomum* spp. eggs shed by male and female calves. EPG, eggs per gram. Bars represent standard error of the mean.

**Figure 4 pathogens-14-01074-f004:**
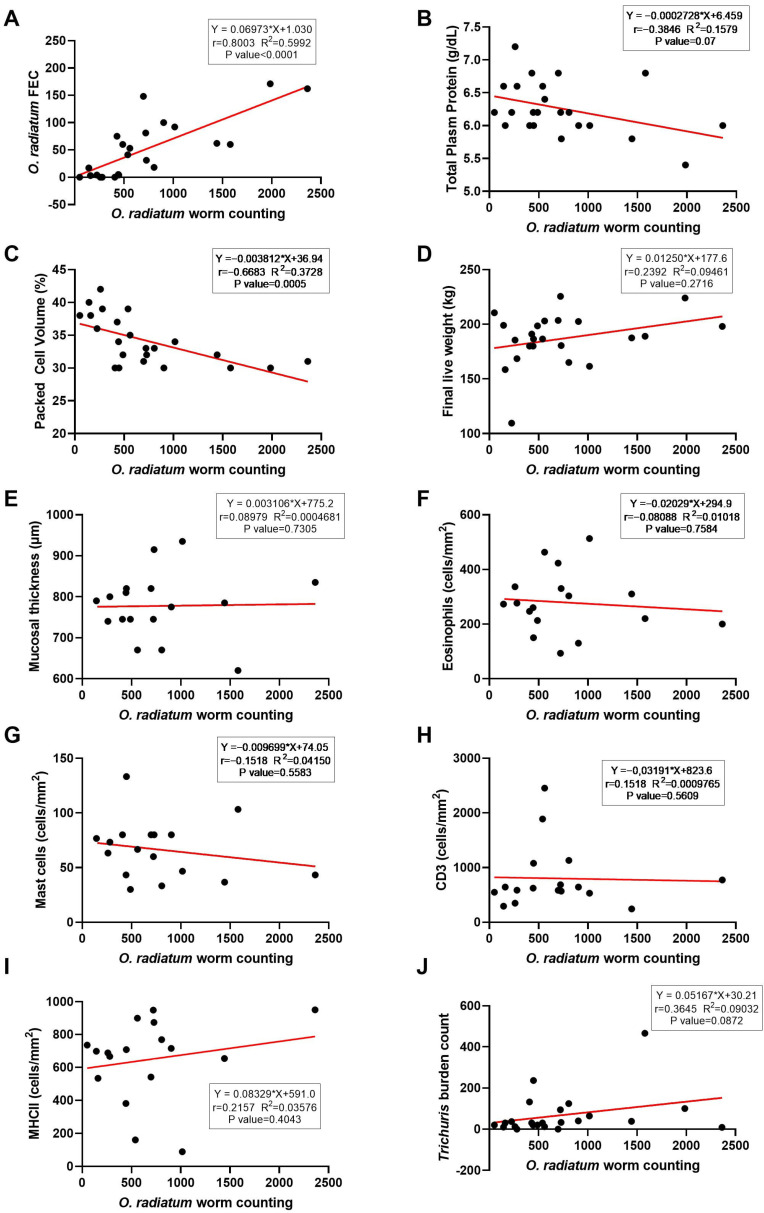
Spearman’s correlation and linear regression analysis of *Oesophagostomum radiatum* worm count and faecal egg count (FEC). (**A**) Total plasma protein (TPP); (**B**) Packed cell volume (PCV); (**C**) Final live weight; (**D**) Mucosal thickness; (**E**) Eosinophils; (**F**) Mast cells; (**G**) CD3+ cells; (**H**) MHC II+ cells; (**I**) *Trichuris* spp.; (**J**) Worm count in naturally infected Nellore calves.

**Figure 5 pathogens-14-01074-f005:**
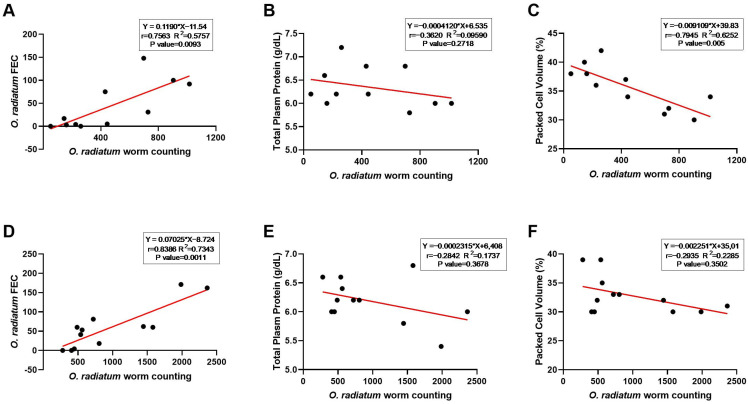
Spearman’s correlation and linear regression analysis of *Oesophagostomum radiatum* worm count and faecal egg count (FEC), total plasma protein (TPP) and packed cell volume (PCV), considering the sex (for females in the panels (**A**–**C**) and for males in the panels (**D**–**F**) of naturally infected Nellore calves.

**Figure 6 pathogens-14-01074-f006:**
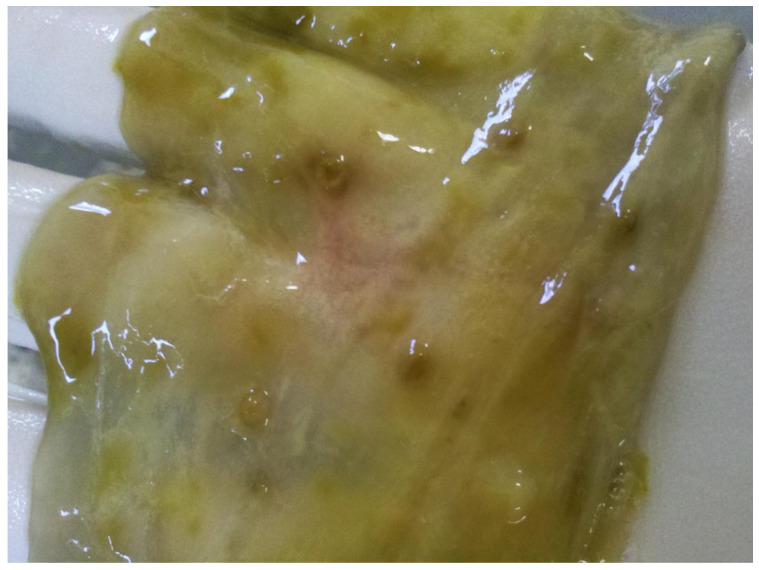
Nodule formation in the large intestine of a calf caused by *Oesophagostomum radiatum* larva infection.

**Figure 7 pathogens-14-01074-f007:**
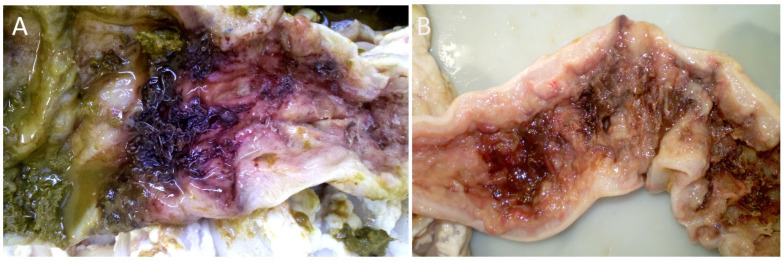
Haemorrhagic lesions in the large intestine of a calf caused by *Oesophagostomum radiatum*. (**A**) Cecocolic junction with a large number of *O. radiatum*; (**B**) Haemorrhagic lesions caused by *O. radiatum* infection along the colon.

**Figure 8 pathogens-14-01074-f008:**
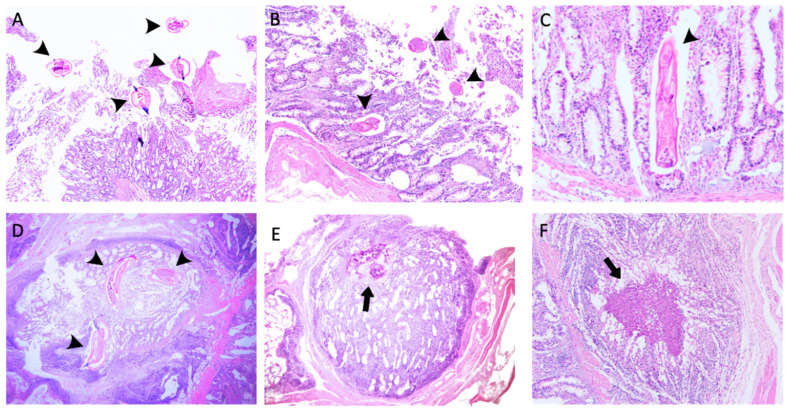
Histopathological changes in the large intestine mucosa of *Oesophagostomum radiatum*-infected calves. (**A**–**C**) Parasites in the mucosa and lumen (arrowhead); (**D**) Parasites inside the submucosa surrounded by inflammatory cells (arrowhead); (**E**) Calcification area (black arrow); (**F**) *Eosinophilic granuloma* (black arrow). (**A**) ×100 magnification; (**B**) ×100 magnification; (**C**) ×200 magnification; (**D**) ×40 magnification; (**E**) ×40 magnification; (**F**) ×100 magnification.

**Figure 9 pathogens-14-01074-f009:**
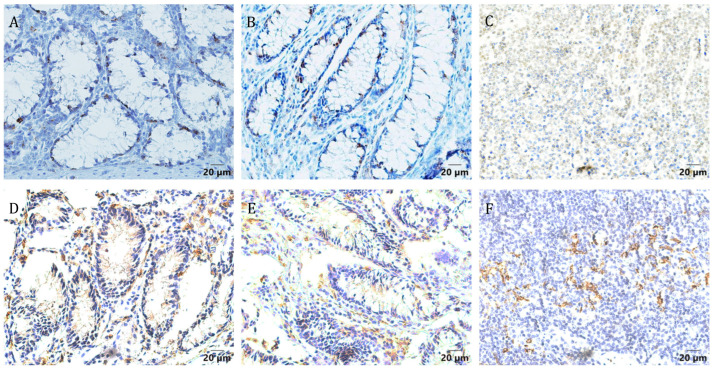
Immunohistochemical labelling of large intestine mucosa sections from *Oesophagostomum radiatum*-infected calves shows positive cells. (**A**–**C**) CD3+ cells; (**D**–**F**) MHC II+ cells (×400 magnification). (**A**,**D**) Intestinal mucosa from female; (**B**,**E**) Intestinal mucosa from male; (**C**,**F**) Lymph node tissue as positive control.

**Table 1 pathogens-14-01074-t001:** Means and standard error (±SE) of variables of 13-month-old calves (*n* = 23) naturally infected by *Oesophagostomum radiatum*. The variables evaluated were the total *O. radiatum* worm count, *Trichuris* spp. count, final live weight (kg), total plasma protein (TPP, g/dL), packed cell volume (PCV, %), eosinophil and mast cell count, CD3+ cells, MHC II+ cells, and mucosal thickness of the large intestine.

**Variables**	**General Mean (±SE)**	**Sex**	**Mean (±SE)**	** *p * ** *****
*O. radiatum* worm count	725 (±124.8)	Male	969 (±200.5)	0.0439 ^$^
		Female	460 (±99.5)	
*Trichuris* count	68 (±21.5)	Male	105 (±38.4)	0.1631 ^$^
		Female	27 (±5.3)	
Final live weight (kg)	187 (±5.1)	Male	193 (±5.4)	0.2271 ^#^
		Female	180 (±8.7)	
TPP (g/dL)	6.3 (±0.1)	Male	6.18 (±0.1)	0.3565 ^#^
		Female	6.35 (±0.1)	
PCV (%)	34.17 (±0.8)	Male	32.83 (±0.9)	0.0712 ^#^
		Female	35.64 (±1.2)	
Eosinophils (cells/mm^2^)	279 (±27.5)	Male	247.7 (±32.0)	0.1812 ^#^
		Female	323.8 (±46.3)	
Mast cells (cells/mm^2^)	67 (±6.5)	Male	66 (±10.5)	0.9339 ^#^
		Female	67 (±6.2)	
CD3+ cells (cells/mm^2^)	802 (±138.9)	Male	829 (±201.5)	0.7582 ^$^
		Female	763 (±193.8)	
MHC II+ cells (cells/mm^2^)	647 (±59.9)	Male	717 (±48.3)	0.1713 ^#^
		Female	548 (±124.3)	
Mucosal thickness (µm)	777.6 (±19.6)	Male	743.5 (±22.4)	0.0326 ^#^
		Female	826.4 (±27.3)	

* *p* value: Means were compared by sex (male and female) by the unpaired *t* test ^#^ or Mann–Whitney test ^$^.

## Data Availability

The data presented in this study are available on request from the corresponding author.

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
