# Peer review of "Immunopathological Changes Caused by *Oesophagostomum radiatum* in Calves: Insights into Host–Parasite Interactions"

_pathogens, 2025, doi:10.3390/pathogens14111074_

Round 1

Reviewer 1 Report

Comments and Suggestions for Authors

This is a well-conducted and valuable study that provides significant insights into the immunopathology of natural Oesophagostomum radiatum infections in Nellore calves. The experimental design is robust, the methodology is comprehensive, and the results are clearly presented. The findings on sex-based differences in worm burden and mucosal responses are particularly noteworthy and constitute a strong contribution to the field. The writing is generally clear, though some specific points require revision to strengthen the manuscript's conclusions and avoid potential misinterpretations.

The central statistical analysis and reporting of the correlation between O. radiatum Faecal Egg Count and total worm burden should be removed from the manuscript. This includes its presence in the Abstract, Statistical Analysis, Results, Discussion, and Conclusions.

There are two main reasons for this, which are the lack of novelty, as the finding that faecal egg count is positively correlated with adult worm burden is a fundamental and well-established principle in parasitology. Its confirmation in this specific system does not provide novel scientific insight.

The second reason is that the O. radiatum FEC was estimated based on the total EPG and the percentage of Oesophagostomum-type L3 from faecal cultures. The strongylid eggs of Oesophagostomum, Haemonchus, Cooperia, are morphologically indistinguishable under the microscope. Therefore, the "O. radiatum FEC" is not a direct measurement but an extrapolation. Presenting a strong correlation (e.g., r=0.80) between this estimated value and the actual worm count can be misleading, as it implies a diagnostic precision that standard coprological techniques do not possess for species-specific quantification in a natural, mixed-breeding setting. This could inadvertently reinforce an unreliable practice in the field.

The manuscript's strengths lie in its direct pathological and immunological observations. Removing this correlative analysis would sharpen the focus on these more novel and impactful results.

Author Response

Point-by-point response to Comments and Suggestions for Authors

Comments 1: This is a well-conducted and valuable study that provides significant insights into the immunopathology of natural Oesophagostomum radiatum infections in Nellore calves. The experimental design is robust, the methodology is comprehensive, and the results are clearly presented. The findings on sex-based differences in worm burden and mucosal responses are particularly noteworthy and constitute a strong contribution to the field. The writing is generally clear, though some specific points require revision to strengthen the manuscript's conclusions and avoid potential misinterpretations.

Response 1: Thank you very much for taking the time to review this manuscript. Please find detailed responses below.

Comments 2: The central statistical analysis and reporting of the correlation between O. radiatum Faecal Egg Count and total worm burden should be removed from the manuscript. This includes its presence in the Abstract, Statistical Analysis, Results, Discussion, and Conclusions.

There are two main reasons for this, which are the lack of novelty, as the finding that faecal egg count is positively correlated with adult worm burden is a fundamental and well-established principle in parasitology. Its confirmation in this specific system does not provide novel scientific insight.

The second reason is that the O. radiatum FEC was estimated based on the total EPG and the percentage of Oesophagostomum-type L3 from faecal cultures. The strongylid eggs of Oesophagostomum, Haemonchus, Cooperia, are morphologically indistinguishable under the microscope. Therefore, the "O. radiatum FEC" is not a direct measurement but an extrapolation. Presenting a strong correlation (e.g., r=0.80) between this estimated value and the actual worm count can be misleading, as it implies a diagnostic precision that standard coprological techniques do not possess for species-specific quantification in a natural, mixed-breeding setting. This could inadvertently reinforce an unreliable practice in the field.

Response 2: Yes, we agree that FEC and worm burden is well-established principles. However, specifically regarding O. radiatum, there are only a few papers on naturally infected Nellore cattle. Therefore, in our opinion, keeping this information in the manuscript is essential to endorse the pathological and immunological observations, facilitating understanding by non-specialists.

We also agree that O. radiatum FEC is an extrapolation, but it is the only way that we can estimate the Oesophagostomum worm burden in the field, and it has been accepted previously, as stated in lines 354-362, and the cited references 17, 30 and 31.

Reviewer 2 Report

Comments and Suggestions for Authors

Overall Assessment: The manuscript is a high-quality, comprehensive, and scientifically sound study that is exceptionally well-written and presented.

It represents a significant contribution to veterinary parasitology. The methodology is robust, and the figures and tables are of high quality. 

Major Suggestion:

Strengthen the Interpretation of Sex-Based Differences: This is the most novel finding. The authors conclude that female calves have a "more effective tissue response" because they have fewer worms and thicker mucosa. The reviewer suggests elaborating on this in the Discussion, as the data showing a negative correlation between worm count and anemia (PCV) only in females could also be interpreted as a more pathologically damaging response.

The authors are encouraged to use their detailed histopathological and immunohistochemical data (e.g., density of CD3+ or MHC II+ cells) to provide more evidence supporting their "more effective response" hypothesis.

Minor Suggestions:

Formalize Clinical Data: The authors mention diarrhea as a main clinical sign in the Discussion. It is suggested that the observational data on fecal consistency scores be formally presented in the Results section to provide quantitative support.  

Improve Legend Consistency: For maximum clarity, all figure and table legends should explicitly define what the error bars represent (e.g., "mean ± SEM"), as this is currently missing in some legends like Figure 3. 

Clarify Figure 8 Legend: There may be a minor error in the legend for Figure 8F, which is described as an "Eosinophilic granuloma" but appears to be a higher magnification of the calcification shown in Figure 8E. The authors should review this for accuracy.

Overall comment

This manuscript represents a significant contribution to the field of veterinary parasitology. The suggestions above are intended to further refine an already strong paper.

Author Response

Point-by-point response to Comments and Suggestions for Authors

Comments 1: The manuscript is a high-quality, comprehensive, and scientifically sound study that is exceptionally well-written and presented.

It represents a significant contribution to veterinary parasitology. The methodology is robust, and the figures and tables are of high quality.

Response 1: Thank you very much for taking the time to review this manuscript.

Thank you very much for your kind words and positive feedback regarding the quality of our manuscript. We truly appreciate your thoughtful comments and encouragement. Your recognition is very motivating and means a lot to us.

Please find detailed responses below and the corresponding revisions/corrections in track changes in the resubmitted file.

Comments 2: Strengthen the Interpretation of Sex-Based Differences: This is the most novel finding. The authors conclude that female calves have a "more effective tissue response" because they have fewer worms and thicker mucosa. The reviewer suggests elaborating on this in the Discussion, as the data showing a negative correlation between worm count and anemia (PCV) only in females could also be interpreted as a more pathologically damaging response.

Response 2: Thank you for pointing this out. We agree that the sex-based interpretation can be further improved. In general, the observed significant negative correlation between O. radiatum worm burden and PCV only in female calves may reflect a stronger but also more energetically costly inflammatory response. We incorporated these considerations in the revised Discussion (lines 355–368).

Comments 3: The authors are encouraged to use their detailed histopathological and immunohistochemical data (e.g., density of CD3+ or MHC II+ cells) to provide more evidence supporting their "more effective response" hypothesis.

Response 3: We have inserted these data in a supplementary table (Table S3).

Comments 4: Formalize Clinical Data: The authors mention diarrhea as a main clinical sign in the Discussion. It is suggested that the observational data on fecal consistency scores be formally presented in the Results section to provide quantitative support.

Response 4: Regrettably, we merely recorded whether the animal exhibited diarrhoea. We did not utilise the scoring scale during the sampling; nevertheless, we could certainly infer it for inclusion in the publication.

Comments 5: Improve Legend Consistency: For maximum clarity, all figure and table legends should explicitly define what the error bars represent (e.g., "mean ± SEM"), as this is currently missing in some legends like Figure 3.

Response 5: Corrected.

Comments 6: Clarify Figure 8 Legend: There may be a minor error in the legend for Figure 8F, which is described as an "Eosinophilic granuloma" but appears to be a higher magnification of the calcification shown in Figure 8E. The authors should review this for accuracy.

Response 6: We have conducted a review, and the legend is accurate. Calcification, at the microscopic level, manifests as the deposition of highly basophilic material, exhibiting an amorphous or granular appearance, frequently organised into clusters. In contrast, an eosinophilic nodule or granuloma manifests as a nodular structure mostly consisting of eosinophils, potentially accompanied by other cell types, including plasma cells, macrophages, or lymphocytes. Due to its association with necrosis, cellular debris may be observed. The granuloma in the cited article is in the exudative-productive phase, exhibiting a more organised structure with a defined circumferential appearance, characterised by a dense collagen fibre composition and specific inflammatory cells at its periphery, according to Malta et al., 2022. (https://doi.org/10.3390/microorganisms10102022).

Comments 7: Overall comment: This manuscript represents a significant contribution to the field of veterinary parasitology. The suggestions above are intended to further refine an already strong paper.

Response 7: Once again, we appreciate your words and suggestions, all of which we have addressed.